## Replications

psychology

post-error slowing, belief in free will, replication, action control

**Author for correspondence:**
Charlotte Eben
e-mail: charlotte.eben@ugent.be

# Are post-error adjustments influenced by beliefs in free will? A failure to replicate Rigoni, Wilquin, Brass and Burle, 2013

Charlotte Eben[1], Zhang Chen[1], Emiel Cracco[1], Marcel Brass[1,2], Joël Billieux[3] and Frederick Verbruggen[1]

[1]Department of Experimental Psychology, Ghent University, Ghent, Belgium
[2]School of Mind and Brain/Department of Psychology, Humboldt Universität zu Berlin, Germany
[3]Institute of Psychology, University of Lausanne, Lausanne, Switzerland

CE, 0000-0001-9423-1261; ZC, 0000-0002-3500-9182;
JB, 0000-0002-7388-6194; FV, 0000-0002-7958-0719

In this pre-registered study, we tried to replicate the study by Rigoni *et al.* 2013 *Cognition* **127**, 264–269. In the original study, the authors manipulated the participants' belief in free will in a between-subject design and subsequently measured post-error slowing (i.e. slower responses after an incorrect trial compared with a correct trial) as a marker of cognitive control. They found less post-error slowing in the group with reduced belief in free will (anti-free will group) compared with a control group in which belief in free will was not manipulated. In the present study, we used the same task procedure and the same free will manipulation (Crick text) in an attempt to replicate these findings. However, we used an online procedure and a larger sample size in order to address concerns about statistical power. Similar to the original study, we also used a questionnaire to measure beliefs in free will as an independent manipulation check. We found a difference in the scores on the questionnaire, thus a reduced belief in free will, after reading the Crick text. However, we did not find any difference in post-error slowing between the anti-free will and control groups. Our findings are in line with several other recent findings suggesting that the Crick text manipulation affects the participants' self-reported belief in free will but not their behaviour. The present study can be considered a high-powered failed replication attempt.

# 1. Introduction

Exerting self-regulation is considered crucial to resist immediate temptations and fulfil long-term goals instead. In the cognitive domain, researchers often refer to 'cognitive control' as part of self-regulatory mechanisms (for a review of various 'regulation' and 'control' concepts in psychology, and how these are related, see [1]). One important aspect of cognitive control is the monitoring of ongoing actions and the adjustment of task parameters and settings when required [1–3]. An often-cited example of action monitoring and subsequent control adjustments is the so-called post-error slowing (PES), which refers to response slowing after an error has been committed. PES has been observed across a wide range of behavioural laboratory tasks, and failures to make adjustments after errors or suboptimal outcomes have been associated with a wide range of psychopathological symptoms and behavioural problems [4,5]. Therefore, PES has become an important marker to study control of flexible and goal-directed behaviour. In the present study, we aimed to replicate a study by Rigoni *et al.* [6], who showed that PES is influenced by the participants' belief in free will.

According to traditional cognitive-control accounts, PES is a consequence of response-threshold adjustments after an error in order to reduce subsequent errors [2,7–9]. However, some studies found that response slowing is not necessarily associated with an increase in response accuracy [10–12]. This finding is inconsistent with the widespread idea that PES is due to cognitive-control adjustments. To reconcile these findings, recent 'adaptive orienting' accounts [9] propose that the slowing after errors can be due to two different mechanisms, depending on the timescale of events. Immediately after the (usually infrequent) error is committed, attention is oriented away from the main task; when the next trial is presented soon after that, responding will be slowed down but accuracy will be *lower* (due to the need to reorient attention) [13]. But when sufficient time is available (i.e. when the interval between consecutive trials is long enough), control adjustments can be made, resulting in longer response latencies and *higher* accuracy.

Interestingly, research suggests that these post-error adjustments can be modulated by individuals' beliefs about their ability to control their own thoughts and actions [14–17]. Belief in free will is an inherent property of the Western moral and social culture, as well as the Western judicial system [18]. However, such beliefs are not static, and might be altered when confronted with e.g. novel information. For example, Rigoni *et al.* [6] manipulated participants' belief in free will, and investigated how this influenced PES. In their experiment, there were two sessions. In the first session (the baseline session), participants filled out a questionnaire assessing their belief in free will (the 'Free Will and Determinism scale plus' or 'FAD plus'; [19]) and then performed a Simon task in which PES was measured. In the second session (which took place a week after the first session), participants were divided into a control group and an anti-free will group. Both groups read a text from *The astonishing hypothesis* by Francis Crick [20]. The control group read a part about consciousness, whereas the anti-free will group read a part stating that scientists found out that free will does not exist [20]. After reading the text (i.e. the experimental manipulation), participants again completed the FAD plus and performed the Simon task in which PES was measured. The pre–post-manipulation differences in PES size and FAD plus scores were calculated, and compared between the groups. Rigoni and colleagues found that reading the anti-free will text reduced both belief in free will (as measured by the FAD plus) and post-error adjustments (as indicated by reduced PES). These results were interpreted as evidence for the idea that reduced belief in free will reduces motivation to exert self-regulation, including performance adjustments after erroneous responses. This finding could have important implications for how societies deal with (perceived) behavioural control problems [21]. For example, certain externalizing behaviours (e.g. attention-deficit/hyperactivity disorder and substance use disorder) are now understood in some societies as neurological disorders, best treated with medications. This raises fundamental questions about volition and responsibility (disease versus lifestyle choice). The findings by Rigoni *et al.* suggest that reduced beliefs about free will (volition and responsibility) can influence further self-regulation. Thus, how societies define (perceived) control problems could influence how individuals self-regulate their behaviour.

In a follow-up study, the same group demonstrated that a neural marker of error monitoring was also influenced by the free will belief manipulation [22]. Specifically, the amplitude of the error-related negativity, a purported neural marker of error detection in electroencephalography (EEG) measurements, was mitigated under the condition of reduced belief in free will. However, the behavioural effect (i.e. reduced PES in the anti-free will group) could not be replicated. Moreover, a new meta-analysis [23] conducted by some of the authors of the current paper indicates that the manipulation of the belief in free will reduces scores on several questionnaires (measuring the belief in free will); however, the overall effect size on free will beliefs is very small, $g = 0.31$ (152 outcomes from 119 studies, calculated using robust variance estimation). This leads to the assumption that the study by Rigoni *et al.* [6] might constitute a false positive result.

Therefore, the aim of this study was to directly replicate the behavioural findings of Rigoni *et al.* [6], as this study suggests that free will beliefs seem to also influence cognitive-control processes.

We pre-registered the entire procedure (https://osf.io/kmpdh) and uploaded all experimental and analyses scripts on OSF (https://osf.io/3nxpg/), and followed the original procedure as much as possible. Therefore, we used the Simon task to measure PES and the Crick [20] text to manipulate belief in free will. In order to address the possible power problem of the original study, we decided to test this replication in a large online sample and follow a Bayesian sampling plan. However, for practical reasons in online testing, we did not have two test sessions as in the original study. Instead, we had only one session that consisted of (i) a baseline phase (in which we measured PES), (ii) the experimental manipulation phase (i.e. reading the text), and (iii) a post-manipulation phase (in which we measured PES and belief in free will). Moreover, we used the free will inventory (FWI) [24] instead of the FAD plus, as the FWI has been established as the preferred scale to measure free will beliefs [25]. We compared FWI scores between groups as a manipulation check.

## 2. Method

### 2.1. Participants

Four hundred participants (200 per group, recruited via Prolific.co) completed the entire online experiment and were included in the analyses (164 females and 4 who preferred to not indicate their gender; age $M = 23.3$ years, s.d. $= 4.8$ years; *range* $= 18-48$). All participants identified themselves as students in order to keep the sample as similar as possible to the original study. Only participants who were able to speak English were allowed to participate. Participants agreed to the consent form before starting the experiment.

In addition to the 400 participants who completed the experiment, an additional 113 participants finished the experiment and got paid but were excluded from data analyses in accordance to the pre-registered exclusion criteria. First, to check whether participants actually read the manipulation text, we included two yes/no-questions at the end of the study (see https://osf.io/3nxpg/); participants who failed to answer both questions correctly and took less than 45 s to read the text were excluded ($N = 55$; 16 in the control group and 39 in the manipulation group). Second, participants with very low accuracy across congruent and incongruent trials in the baseline Simon task (more than 2.5 standard deviations (s.d.) below the overall sample average in the respective condition) were excluded ($N = 29$). Third, we excluded 21 participants with missing or incomplete datasets due to technical problems (i.e. data were not recorded, probably related to poor Internet connection). Fourth, we excluded seven participants who did not have enough trials for all conditions (see below for further information on trial exclusion criteria). In addition to the pre-registered criteria, we excluded one participant who used the wrong response keys during the whole experiment. In order to reach the predetermined sample size and Bayes factors, all excluded participants were replaced. This data-replacement procedure was pre-registered. Five additional datasets were rejected (not paid and therefore immediately replaced) due to low effort responses (accuracy at guessing level ($\approx 50\%$) and very short response time (less than 100 ms)) and too short completion times. Finally, an additional 109 participants signed up for the experiment but never started or completed it. This seems to be a common procedure on Prolific in order to 'reserve' a spot in a better paid experiment. All included datasets, as well as the excluded datasets and the five rejected datasets can be found on the OSF (https://osf.io/3nxpg/). In this OSF repository, we also included a more detailed explanation of the Bayesian sampling and the replacement procedures.

All data were acquired between 3 February and 5 March 2020. Participants had to enter their nationality manually, as this ensured that no bots could complete the task.

### 2.2. Sample size

We set a minimum sample size of 100 participants per group and a maximal sample size of 200 participants per group (so 400 participants in total). To determine our sample size, we used sequential Bayesian hypothesis testing (see Analyses section for further details) by increasing the sample in steps of 50 participants (25 per group) until a decisive Bayes factor (BF larger than 10 or smaller than 1/10) was obtained [26], or until we reached the maximum number of 400 participants. We calculated the BF based on the ANOVA with test phase as within-subject factor and group as between-subject factor. The difference score between pre- and post-error reaction time as calculated with the Dutilh *et al.* [27] method (see below) was our main dependent variable, and therefore, used to determine the sample size. This measure is not contaminated by global fluctuations in response

latency, unlike the traditional method used in the original study. The minimum sample size (100 per group or 200 in total) was based on recommendations by Brysbaert [28] for a standard two-way ANOVA with one within-subject factor, one between-subject factor, and with $r = 0.90$ (i.e. the correlation between the within-subject variables) [28].

## 2.3. Apparatus and stimuli

The experiment was programmed in jsPsych (v. 6.0.5), which has a comparable accuracy in response times measurements to standard lab software like Psychophysics Toolbox and E-Prime [29]. The experiment only ran on desktop computers and laptops, with Chrome or Mozilla Firefox installed (the experiment usually runs without any problems in these two browsers [29]). Keyboards were used to register responses. Participants had to respond to the colour of a circle (blue or yellow) by pressing a left or right key. We kept the S-R mapping the same across all participants with the 'r' key assigned to the yellow circle and the 'p' key assigned to the blue circle. Note that the colours in the original study were green and red, but in order to avoid problems for colour blind participants, we used blue and yellow here. The circles appeared either on the left or the right side of the screen, but the stimulus location was irrelevant. Therefore, there were trials in which the response key and the location of the stimulus matched (congruent trials; i.e. left side on the screen and left response key, or right side on the screen and right response key), and trials on which the response key and location did not match (incongruent trials; i.e. left side of the screen and right response key, or right side on the screen and left response key). The colour and stimulus location were pseudo-randomized within subjects (equal probability of all combinations; i.e. yellow left location, yellow right location, blue left location and blue right location).

## 2.4. Procedure

The general procedure started with welcoming the participant, asking for consent, requesting information about age, gender (with the options 'male', 'female' and 'I don't want to say') and nationality, and explaining the experimental procedure. The proper experiment consisted of three main phases.

In the first phase (the baseline phase), participants performed the Simon task. In accordance with the original code provided by the first author of the original study [6], first a fixation cross was presented for 500 ms, after which the stimulus (a blue- or yellow-coloured circle) was presented. The stimulus remained on the screen until the participant's response. The next trial started immediately after the response (i.e. there was no feedback or extra inter-trial interval). In total, participants completed a practice block with 16 trials and four experimental blocks with 96 trials each. We tested this Simon task procedure in a pilot study ($N = 40$) and observed the expected PES (Hedge's $g_{av} = 1.6$). The data and the code of this pilot experiment are made available as well (https://osf.io/3nxpg).

In the second phase (the manipulation phase), participants read a passage from *The astonishing hypothesis*, by Francis Crick [20]. Which passage they had to read depended on the group they were assigned to: the anti-free will group read a text claiming that scientists now recognize that free will is an illusion, while the control group read a passage from the same book that did not mention free will. Participants were instructed that they had to answer two questions about the text at the end of the experiment. This was to ensure that participants read the manipulation text carefully. The texts and the questions are made available on OSF (https://osf.io/3nxpg/). We had to run the two groups consecutively (first the control group; then the anti-free will group) due to technical restrictions by Prolific to ensure that participants could not participate twice (i.e. once as part of the control group and once as part of the anti-free will group).

In the third phase (the post-manipulation phase), participants again performed four blocks of the Simon task with 96 trials each. When all blocks were completed, participants filled in the FWI and an impulsivity questionnaire (UPPS-P short). The FWI (see https://osf.io/3nxpg/) consists of 15 Likert-type items (scores from 1 = *strongly disagree* to 7 = *strongly agree*) with three subscales (Free Will, Determinism and Dualism/Anti-Reductionism). The impulsivity questionnaire was an addition, as this project will be part of a larger individual-differences project across studies. We did not use the UPPS-P data for this study.

Thus, the overall procedure is highly similar to the procedure used in the original study [6]. There are only four notable procedural differences. First, this study was conducted online to reach a larger sample size. To our knowledge, there are at least two published studies [25,30] which used the Crick text in a single online procedure and both studies observed a difference between the anti-free will group and the control group in their manipulation checks. Moreover, the meta-analysis mentioned above [23]

included these two studies and other unpublished studies to compare the Crick-text manipulation in laboratory-based and online contexts. The meta-analysis found a small effect in the laboratory studies, ($g = 0.21$, s.e. $= 0.04$, $p < 0.001$, based on 25 effect sizes from 24 experiments), as well as a small effect in the online studies ($g = 0.13$, s.e. $= 0.05$, $p < 0.012$, based on 22 effect sizes from 20 experiments). The numerical difference between laboratory-based and online studies did not reach significance ($p = 0.21$). We, therefore, assume that this first deviation from the original study protocol (i.e. online instead of laboratory-based) did not affect the results negatively; instead, we assume it improved the design as we were able to test a much larger (and potentially more heterogeneous) sample. Second, we used the FWI instead of the FAD plus (which was used in the original study), because the FWI might be more sensitive than the FAD plus [25]. For this reason, other follow-up studies have also used this FWI. Third, our study consisted of only one session, whereas the original study [6] consisted of two distinct test sessions that were separated by a week. However, we only had one test session for practical reasons as testing up to 400 participants or more (depending on how many participants had to be replaced) across two online sessions did not seem feasible. Because of this, we also decided to drop the baseline FWI measurement. If we would have included a baseline FWI measurement, the baseline and post-manipulation measurement of the FWI would have been separated by only 30 min, making carry-over effects likely. However, note that our design was sufficiently powered to detect between-subject differences in FWI scores (resulting from the experimental manipulation) [28]. Fourth, we also decided to drop the PANAS, which was used in the original study to measure positive and negative affect, as Rigoni *et al.* found no effect of reading the manipulation text on the PANAS scores.

## 2.5. Analyses

We conducted all analyses as they were conducted in the original study and in line with our pre-registered data-analytic plan (https://osf.io/kmpdh). All data processing and analyses were completed with R (v. 3.6.2) using the packages here (v. 0.1), ez (v. 4.4-0), Hmisc (v. 4.3-1), ggplot (v. 3.2.1), doBy (v. 4.6-4.1), BF (v. 0.9.12-4.2) and tidyverse (v. 1.3.0) [31]. All R and JASP scripts of the experiment (including pre-registered scripts and scripts for the additional, exploratory analyses) can be found on OSF (https://osf.io/3nxpg/).

### 2.5.1. Data processing

For each trial type (post-correct versus post-error), we calculated how fast participants responded to the stimulus (response time [RT]). RTs shorter than 100 ms and longer than 1 s were not considered for the analysis. Moreover, missing trials (due to online data collection) and the following trial were excluded as we could not determine for the missing trials whether the responses were correct or not. The trial exclusion criteria were determined before data collection and matched the exclusion criteria used by Rigoni *et al.* [6]. The number of trials excluded depended on how we calculated PES.

We measured PES in two distinct ways: the traditional difference score (which was used in the original study) and the Dutilh method (which is less biased by global fluctuations). First, we calculated a difference score between RTs on trials following an error and RTs on trials following a correct trial. In the original study, this (often-used) difference score is referred to as the 'PES effect'. Second, we measured PES as suggested by Dutilh *et al.* [27], focusing on 'correct–**correct**–error–**correct**' trial sequences (figure 1). In such sequences, the trial before the error is a post-correct trial and the trial after the error is a post-error trial. We calculated the difference between these trials for each of these trial sequences and then calculated the mean difference score. This difference provides a measure of PES that is no longer contaminated by global fluctuations [27].

### 2.5.2. Data analysis

The main analyses focused on PES as a function of the experimental free will manipulation. For the Dutilh *et al.* [27] method, we used the PES measure as explained above and submitted it to an ANOVA with test phase (baseline versus post-manipulation) as within-subject factor and group (anti-free will versus control) as between-subject factor. In case of a significant interaction, we used paired Bayesian *t*-tests to compare the PES difference across sessions between the groups in order to get an analytical solution for the BF (figure 1).

For the traditional analyses of PES (as used in the original study as well), RTs and error rates were submitted to an ANOVA (frequentist and Bayesian equivalent) with test phase (baseline versus

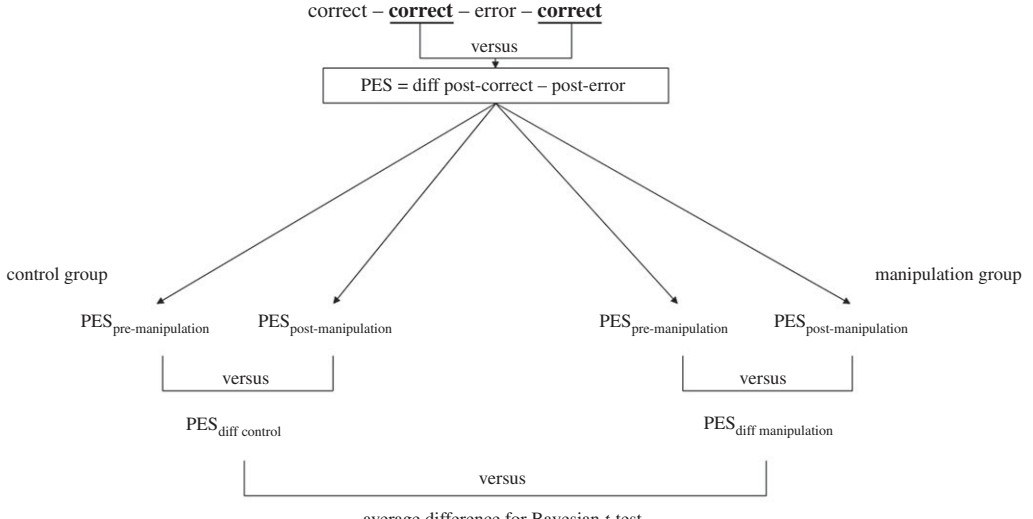

**Figure 1.** Schematic overview of the Dutilh *et al.* [27] method to calculate post-error slowing and the planned comparisons.

post-manipulation) and previous trial (correct or error) as within-subject factors, and group as between-subject factors. We used paired *t*-tests (frequentist and Bayesian equivalent) to measure the change of PES across sessions within each group. For this, we calculated a difference score between trials following errors and trials following correct responses and then the difference between the baseline and the post-manipulation phase in this difference score for each group.

To check whether there were already differences between the groups in the baseline phase, we also submitted the mean correct RTs and the mean error rate in the baseline phase to a mixed ANOVA (frequentist and Bayesian equivalent) with congruency (congruent, incongruent) as within-subjects factor and group (anti-free will, control) as between-subjects factor. Finally, for the extra manipulation check, we ran a *t*-test (frequentist and Bayesian equivalent) for independent samples on the FWI subscale scores to compare the score between the groups (anti-free will, control).

We also pre-registered an additional exploratory analysis concerning congruency as an additional factor. However, after more consideration we came to the conclusion that there would be not enough trials for this analysis. Therefore, in contradiction to the pre-registration, we did not perform this analysis. But as discussed below, we did perform two different exploratory analyses (that were not pre-registered).

We report *p*-values and BFs. All tests (frequentist and Bayesian) were two-tailed. As this is a replication, a one-tailed *t*-test might be viewed as more appropriate. However, we had concerns about the original (low) sample size, and as such, deemed it possible that the effect could also go in the opposite direction in our replication. Moreover, such a finding would still be theoretically meaningful. Therefore, we decided to use two-tailed *t*-tests. For frequentist statistics, significance was determined at an alpha level of 0.05 (but with correction for multiple comparisons). BFs in a Bayesian *t*-test ($BF_{10}$) were calculated with the BF package v. 0.9.12-2 in R, and quantified the evidence for the alternative hypothesis against the null hypothesis. In contrast to the pre-registration, in which we said we would perform all analyses in R, we used JASP (v. 0.13.1.0) [32] to perform the Bayesian ANOVA. We report the inclusion $BF_{10}$ of the Bayesian ANOVAs (calculated in JASP) in tables 1–7. The $BF_{10}$ of the Bayesian *t*-tests (calculated in R) are reported in the main text. An inclusion $BF_{10}$ of B means the data are about B times more likely under the models that include this effect, compared to the models that do not include this effect. Given the small sample size of the original study, we assumed that the reported effect size was not a good estimate for the population effect size. Therefore, we decided to use the default prior widths (the Cauchy prior width of 0.707) as defined by the BF package in R and in JASP.

# 3. Results

## 3.1. Belief in free will

We analysed the scores of the three subscales of the FWI separately. Note that high scores on the Free Will subscale (five items) and the Dualism/Anti-Reductionism subscale (five items) meant a high belief in

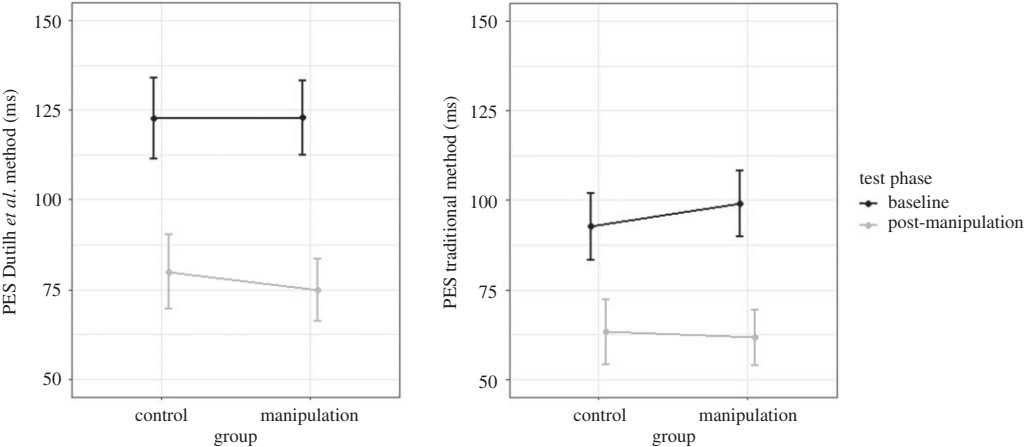

**Figure 2.** PES (in ms) calculated with the Dutilh *et al.* method and the traditional method of PES as a function of group and test phase. The error bars reflect between-subject confidence interval.

**Table 1.** Inferential statistics of the Dutilh *et al.* [27] difference score.

| effect | DFn | DFd | MSE | F | p-value | p < 0.05 | ges | BF$_{10}$ |
|---|---|---|---|---|---|---|---|---|
| group | 1.00 | 398.00 | 7965.04 | 0.16 | 0.69 | | 0.00 | 0.143 |
| test phase | 1.00 | 398.00 | 2836.30 | 145.72 | 0.00 | * | 0.09 | >100 |
| group × test phase | 1.00 | 398.00 | 2836.30 | 0.47 | 0.49 | | 0.00 | 0.129 |

Note: ges, generalized eta-squared measure of effect size.

free will, whereas high scores on the Determinism subscale (five items) meant a low belief in free will. For the manipulation check, we ran Welch's *t*-tests for independent samples on the FWI subscale scores to compare the score between the groups (anti-free will, control). For the Determinism subscale, the *t*-test revealed a significant difference between the anti-free will group ($M = 3.75$; s.d. = 1.15) and the control group ($M = 3.52$; s.d. = 1.12), $t(397.74) = 1.99$, $p = 0.05$, $g_{av} = 0.20$. However, the corresponding BF was indecisive, BF$_{10} = 0.74$. For the Dualism/Anti-Reductionism subscale, the *t*-test revealed a significant difference between the anti-free will group ($M = 3.87$; s.d. = 1.40) and the control group ($M = 4.33$; s.d. = 1.42), $t(397.87) = -3.27$, $p < 0.001$, $g_{av} = 0.33$. The corresponding BF showed strong evidence for the alternative hypothesis, BF$_{10} = 18.04$. For the Free Will subscale the *t*-test revealed a significant difference between the anti-free will group ($M = 4.25$; s.d. = 1.18) and the control group ($M = 4.58$; s.d. = 1.20), $t(397.95) = -2.75$, $p < 0.001$, $g_{av} = 0.28$. The corresponding BF showed moderate evidence for the alternative hypothesis, BF$_{10} = 4.19$.

## 3.2. Post-error slowing in the Simon task

### 3.2.1. Dutilh *et al.* [27] method

We used the Dutilh *et al.* [27] difference score and submitted it to an ANOVA with test phase (baseline versus post-manipulation) as within-subject factor and group (anti-free will versus control) as between-subject factor. The difference scores are presented in figure 2 and the inferential statistics are presented in table 1. The ANOVA revealed a significant main effect of test phase, indicating a larger difference score in the baseline phase ($M = 123$ ms; s.d. = 78 ms) than in the post-manipulation phase ($M = 78$ ms; s.d. = 69 ms). The main effect of group, and the interaction between test phase and group, were not significant. The independent Bayesian *t*-test to compare the PES differences across test phases between the groups (figure 1), revealed substantial evidence for the null hypothesis, BF$_{10} = 0.139$. Thus, we could not replicate the main finding of Rigoni *et al.* [6] using a PES measure that is not contaminated by global fluctuations.

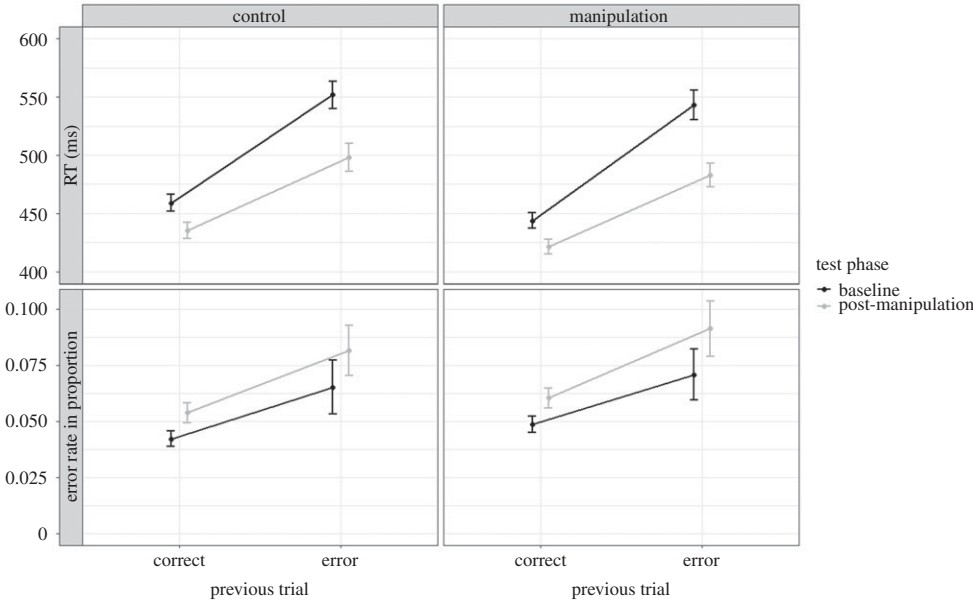

**Figure 3.** RT and error rate in the traditional way of calculating PES as a function of previous trial, group and test phase. The error bars reflect between-subject confidence interval.

**Table 2.** Inferential statistics of the RT data calculated with the traditional measurement of post-error slowing.

| effect | DFn | DFd | MSE | F | p-value | p < 0.05 | ges | BF$_{10}$ |
|---|---|---|---|---|---|---|---|---|
| group | 1.00 | 398.00 | 13 320.40 | 5.28 | 0.02 | * | 0.01 | 1.794 |
| test phase | 1.00 | 398.00 | 1643.30 | 387.39 | 0.00 | * | 0.08 | >100 |
| previous trial | 1.00 | 398.00 | 2971.57 | 843.04 | 0.00 | * | 0.25 | >100 |
| group × test phase | 1.00 | 398.00 | 1643.30 | 0.39 | 0.53 | | 0.00 | 0.092 |
| group × previous trial | 1.00 | 398.00 | 2971.57 | 0.20 | 0.65 | | 0.00 | 0.090 |
| test phase × previous trial | 1.00 | 398.00 | 1064.56 | 104.81 | 0.00 | * | 0.01 | >100 |
| group × test phase × previous trial | 1.00 | 398.00 | 1064.56 | 1.47 | 0.23 | | 0.00 | 0.174 |

Note: ges, generalized eta-squared measure of effect size.

### 3.2.2. Traditional analyses

RTs and error rates were submitted to an ANOVA with test phase (baseline versus post-manipulation) and previous trial (correct versus error) as within-subjects factors, and group as between-subjects factor. PES difference scores are shown in figure 2. For the RT, the ANOVA revealed a significant main effect of group, indicating shorter RTs for the anti-free will group ($M = 473$ ms; s.d. = 81 ms) compared to the control group ($M = 486$ ms; s.d. = 83 ms). The main effect of test phase was also significant, indicating shorter RTs in the post-manipulation phase ($M = 460$ ms; s.d. = 73 ms) than in the baseline phase ($M = 499$ ms; s.d. = 86 ms). The main effect of previous trial was also significant, indicating shorter RTs for trials following a correct trial ($M = 440$ ms; s.d. = 51 ms) compared to trials following an error trial ($M = 519$ ms; s.d. = 89 ms), resulting in a general PES effect of 79 ms. The interaction between test phase and previous trial was significant. Using the difference between trials following a correct trial and trials following an error (PES effect; figure 2), a paired (Bayesian) $t$-test showed that the PES effect was larger in the baseline phase ($M = 96$ ms; s.d. = 66 ms) than in the post-manipulation phase ($M = 62$ ms; s.d. = 60 ms), $t(399) = 10.23$, $p < 0.001$, BF$_{10} > 100$. All other interactions were not significant. For detailed inferential statistics, see table 2 and for detailed descriptive data (figure 3).

For the error rates, the ANOVA revealed no significant main effect of group. The main effect of test phase was significant, indicating higher error rates in the post-manipulation phase ($M = 7.2\%$;

**Table 3.** Inferential statistics of the error data calculated with the traditional measurement of post-error slowing.

| effect | DFn | DFd | MSE | F | p-value | p < 0.05 | ges | BF$_{10}$ |
|---|---|---|---|---|---|---|---|---|
| group | 1.00 | 398.00 | 0.01 | 2.97 | 0.09 | | 0.00 | 0.337 |
| test phase | 1.00 | 398.00 | 0.00 | 33.51 | 0.00 | * | 0.01 | >100 |
| previous trial | 1.00 | 398.00 | 0.00 | 65.13 | 0.00 | * | 0.04 | >100 |
| group × test phase | 1.00 | 398.00 | 0.00 | 0.17 | 0.68 | | 0.00 | 0.103 |
| group × previous trial | 1.00 | 398.00 | 0.00 | 0.03 | 0.86 | | 0.00 | 0.091 |
| test phase × previous trial | 1.00 | 398.00 | 0.00 | 1.87 | 0.17 | | 0.00 | 0.170 |
| group × test phase × previous trial | 1.00 | 398.00 | 0.00 | 0.17 | 0.68 | | 0.00 | 0.114 |

Note: ges, generalized eta-squared measure of effect size.

**Table 4.** Inferential statistics of the RT data the baseline phase.

| effect | DFn | DFd | MSE | F | p-value | p < 0.05 | ges | BF$_{10}$ |
|---|---|---|---|---|---|---|---|---|
| group | 1.00 | 398.00 | 5006.86 | 8.32 | 0.00 | * | 0.02 | 8.092 |
| congruency | 1.00 | 398.00 | 144.29 | 1042.22 | 0.00 | * | 0.07 | >100 |
| group × congruency | 1.00 | 398.00 | 144.29 | 0.62 | 0.43 | | 0.00 | 0.133 |

Note: ges, generalized eta-squared measure of effect size.

s.d. = 6.6%) than in the baseline phase ($M$ = 5.7%; s.d. = 6.3%). The main effect of previous trial was also significant, indicating higher error rates for trials following an error trial ($M$ = 7.7%; s.d. = 8.5%) compared to trials following a correct trial ($M$ = 5.1%; s.d. = 3.0%). All interactions were non-significant. See table 3, for detailed inferential statistics.

## 3.3. Baseline check

We submitted the mean correct RTs and the mean error rate in the baseline phase to a mixed ANOVA with congruency (congruent, incongruent) as within-subjects factor and group (anti-free will, control) as between-subjects factor to check whether the groups already differed from each other before the experimental manipulation. For the RT, the ANOVA revealed a main effect of group, indicating faster RT for the anti-free will group ($M$ = 448 ms; s.d. = 51 ms) compared to the control group ($M$ = 463 ms; s.d. = 54 ms). We tested whether this effect might be related to the exclusion of participants and included all complete datasets. We also found this main effect of group in the analysis with all datasets included, which indicates that this might be rather a false positive result than a sampling bias. The main effect of congruency was also significant, indicating faster RT for congruent trials ($M$ = 442 ms; s.d. = 51 ms) compared to the incongruent trials ($M$ = 469 ms; s.d. = 52 ms). The interaction between group and congruency was not significant. See table 4, for detailed inferential statistics.

For the error rates, the ANOVA also revealed a main effect of group, indicating higher error rate for the anti-free will group ($M$ = 5.0%; s.d. = 3.8%) compared to the control group ($M$ = 4.3%; s.d. = 3.4%). The main effect of congruency was also significant, indicating higher error rates for incongruent trials ($M$ = 6.4%; s.d. = 4.0%) compared to the congruent trials ($M$ = 2.9%; s.d. = 2.0%). The interaction between group and congruency was also significant, indicating a larger congruency effect (incongreunt–congruent) in the manipulation group (3.9%) compared to the control group (3.1%) (table 5).

## 3.4. Not-pre-registered exploratory analyses

Originally, we pre-registered exploratory analyses to test the idea that post-error slowing is modulated by congruency, performing additional analyses with congruency as a third within-subject factor. After some consideration, we decided that there were not enough trials for these analyses including all within- and

**Table 5.** Inferential statistics of the error data the baseline phase.

| effect | DFn | DFd | MSE | F | p-value | p < 0.05 | ges | BF$_{10}$ |
|---|---|---|---|---|---|---|---|---|
| group | 1.00 | 398.00 | 0.00 | 6.47 | 0.01 | * | 0.01 | 2.629 |
| congruency | 1.00 | 398.00 | 0.00 | 359.50 | 0.00 | * | 0.23 | >100 |
| group × congruency | 1.00 | 398.00 | 0.00 | 4.34 | 0.04 | * | 0.00 | 0.826 |

Note: ges, generalized eta-squared measure of effect size.

**Table 6.** Inferential statistics of the exploratory RT analyses including congruency.

| effect | DFn | DFd | MSE | F | p-value | p < 0.05 | ges | BF$_{10}$ |
|---|---|---|---|---|---|---|---|---|
| group | 1.00 | 398.00 | 8938.81 | 8.44 | 0.00 | * | 0.02 | 8.313 |
| congruency | 1.00 | 398.00 | 193.51 | 1365.13 | 0.00 | * | 0.06 | >100 |
| test phase | 1.00 | 398.00 | 615.20 | 368.59 | 0.00 | * | 0.05 | >100 |
| group × congruency | 1.00 | 398.00 | 193.51 | 1.38 | 0.24 | | 0.00 | 0.121 |
| group × test phase | 1.00 | 398.00 | 615.20 | 0.31 | 0.58 | | 0.00 | 0.113 |
| congruency × test phase | 1.00 | 398.00 | 64.43 | 18.43 | 0.00 | * | 0.00 | 0.590 |
| group × congruency × test phase | 1.00 | 398.00 | 64.43 | 0.13 | 0.71 | | 0.00 | 0.081 |

Note: ges, generalized eta-squared measure of effect size.

**Table 7.** Inferential statistics of the exploratory error analyses including congruency.

| effect | DFn | DFd | MSE | F | p-value | p < 0.05 | ges | BF$_{10}$ |
|---|---|---|---|---|---|---|---|---|
| group | 1.00 | 398.00 | 0.00 | 6.15 | 0.01 | * | 0.01 | 2.173 |
| congruency | 1.00 | 398.00 | 0.00 | 445.52 | 0.00 | * | 0.21 | >100 |
| test phase | 1.00 | 398.00 | 0.00 | 124.31 | 0.00 | * | 0.03 | >100 |
| group × congruency | 1.00 | 398.00 | 0.00 | 2.16 | 0.14 | | 0.00 | 0.547 |
| group × test phase | 1.00 | 398.00 | 0.00 | 0.07 | 0.80 | | 0.00 | 0.086 |
| congruency × test phase | 1.00 | 398.00 | 0.00 | 3.03 | 0.08 | | 0.00 | 0.141 |
| group × congruency × test phase | 1.00 | 398.00 | 0.00 | 2.51 | 0.11 | | 0.00 | 0.150 |

Note: ges, generalized eta-squared measure of effect size.

between-subject factors. Therefore, we conducted an additional, non-pre-registered $2 \times 2 \times 2$ exploratory analysis with the independent variables group (control versus anti-free will), test phase (baseline versus post-manipulation) and congruency (congruent versus incongruent), omitting the within-subject factor previous trial, to test whether the free-will manipulation has an influence on interference control (measured via the congruency effects). Whereas all three main effects were significant in RT data as well as error data, we only found an interaction between congruency and test phase in the RT data. None of the interactions with group reached significance (see tables 6 and 7 for detailed inferential statistics and see figure 4 for descriptive data).

Originally, we pre-registered the inclusion of participants who either answered both yes/no questions about the manipulation text correctly at the end of the experiment, or read the text for at least 45 s. More precisely, according to the pre-registration only participants that failed both attention checks were excluded. However, even without reading the text, there is a 25% chance that both questions are answered correctly. In the main analyses, we included 34 participants (23 in the control group and 11 in the anti-free will group) who took less than 45 s to read the text. These may not have read the text properly, and simply guessed correctly at the end of the experiment. Therefore, we ran the same

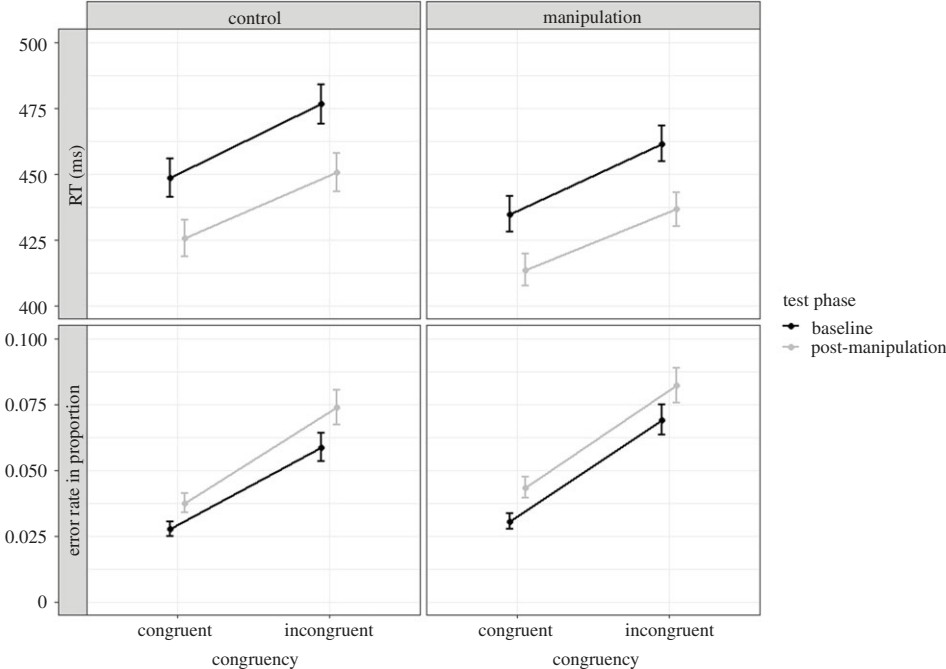

**Figure 4.** RT (in ms) and error rates (in proportion) as a function of congruency and group in the baseline phase and in the post-manipulation phase. The error bars reflect between-subject confidence interval.

analyses again but excluded these 'fast readers'. The analyses revealed the same effects as the pre-registered analyses in the behavioural data, and there were no meaningful changes in inferential statistics. See the OSF repository (https://osf.io/3nxpg/) for more detailed information. By contrast, the analysis of the questionnaire data was affected. In the frequentist analyses, all group differences remained significant (all $ps < 0.05$), but the BFs changed more substantially. The BF for the Determinism subscale remained inconclusive $BF_{10} = 0.88$, whereas the BF for the Dualism subscale decreased and only showed moderate evidence for the alternative hypothesis, $BF_{10} = 6.39$. The BF for the Free Will subscale also decreased, showing only anecdotal evidence for the alternative hypothesis, $BF_{10} = 2.18$.

# 4. General discussion

In this study, we attempted to replicate the influence of free will beliefs on cognitive control. Rigoni *et al.* [6] found that after reading an 'anti-free will text', participants' belief in free will was reduced, and they showed less slowing after an error (an often-used behavioural marker of cognitive-control adjustment). In the anti-free will group, reduction in free-will beliefs also correlated with a reduction in PES. Based on these findings, the authors concluded that weakening beliefs in free will can have an impact on cognitive control (and behaviour more generally). Here, we attempted to replicate their findings. To this end, we used the same Simon task as the original study to measure PES. We also used the Crick [20] text to manipulate the participants' belief in free will. The FWI questionnaire was included as an independent manipulation check. We found an influence of the manipulation on the FWI subscales, with a similar effect size as in the original study. The Cohen's *d* in our study ranged from 0.20 to 0.33 for the different subscales; in the original study, the effect size of the aggregated score of the FAD plus was Cohen's *d* = 0.22. This suggests that the manipulation worked equally well in both studies. Crucially, we failed to replicate the finding that PES in a Simon task was reduced when belief in free will is mitigated. Additional analyses also indicated that the Crick manipulation did not influence behavioural performance. Thus, our results suggest that reduced belief in free will has no impact on behavioural performance after all.

It could be argued that we did not find an effect of free will beliefs on PES because the inter-trial interval (ITI) in our task was too short. As noted in the Introduction, alternative accounts of PES have been proposed. Specifically, it has been argued that errors may orient attention away from the main task because these are rare events. Such attentional reorienting would also increase reaction times in

an initial phase. Only when the ITI is long enough, control adjustments would contribute to the slowing. The higher error rates found after incorrect trials (see the Traditional analyses section) are consistent with this idea. Nevertheless, it is worth noting that the same ITI was used in the original study of Rigoni *et al.* [6], and all behavioural measures were similar (i.e. overall RTs and error rates were comparable across studies). Thus, if PES in the present study was due to attentional reorienting (rather than control adjustments), then this was most likely also the case in the original study. Second, even a reorienting account could predict reduced PES when beliefs in free will are reduced. Specifically, the orienting account predicts that slowing is observed after salient events. But when beliefs in free will are reduced, participants would generally feel less responsible for the outcome of a trial. Consequently, errors would become less salient and PES should be reduced. Thus, it can be argued that PES should be reduced, regardless of the ITI and the relative contribution of reorienting and cognitive-control adjustments to the effect. Third, the exploratory analyses revealed that the congruency effect, which has been used as a marker of cognitive control as well, was similarly not influenced by the free will manipulation. Finally (and perhaps most importantly), a possible alternative interpretation of PES does not alter the observation that we were not able to replicate the effect of the experimental manipulation on PES and behavioural performance in the first place (despite relying on the same design).

Thus, we can conclude that our findings are not consistent with the original study, which supported an influence of free will manipulations on a behavioural measure indexing PES. Our findings are aligned with a recent study which similarly found altered belief in free will in judges after reading the Crick text, but no influence on their subsequent sentencing (i.e. behaviour) [33]. Yet, some recent studies even failed to show an effect on self-reported measures. Several replication attempts investigating the influence of belief in free will on cheating behaviour did not find any significant difference between the anti-free will group and the control group, neither in the self-reported belief in free will measure, nor in the actual cheating behaviour [34,35]. Generally, it seems that if the free-will manipulation 'works', its effect is restricted to self-reported beliefs and does not influence actual behaviour.

# 5. Conclusion

In summary, in this high-powered replication attempt, we were able to find a difference between the anti-free will group and the control group on the questionnaire scores, showing reduced belief in free will in the anti-free will group. This is in line with the original study and a recent meta-analysis [23]. However, we were not able to reproduce the finding that reduced belief in free will leads to reduced PES. Even though we did not reach our aimed BF of larger than 10 or smaller than 0.1, our BF revealed substantial evidence in favour of the null hypothesis. We, therefore, were not able to replicate the central finding that the Crick manipulation of belief in free will influences post-error adjustment behaviour.

Ethics. The study was approved by the local research ethics committee at the Faculty of Psychology and Educational Science of Ghent University. Informed consent was obtained.

Data accessibility. All experimental and analyses codes and materials can be found on https://osf.io/3nxpg/.

Authors' contributions. F.V. and C.E. developed the idea of replication. All authors contributed to the online design of the study. C.E. and Z.C. programmed the experiment. C.E. and F.V. wrote the first draft of the pre-registration and the manuscript. Z.C., E.C., M.B. and J.B. provided critical revisions on the pre-registration and manuscript. Note that Z.C. acted as 'co-pilot' for the whole project (see https://fredvbrug.github.io/openScience.html and [36] for more information).

Competing interests. We declare we have no competing interests.

Funding. This work was supported by an ERC Consolidator grant awarded to F.V. (European Union's Horizon 2020 research and innovation programme, grant agreement no. 769595). M.B. is supported by an Einstein Strategic Professorship of the Einstein Foundation Berlin.

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
