## [Reviewer comments · Royal Society Open Science]

Review History

RSOS-200664.R0 (Original submission)

Review form: Reviewer 1

Do you have any ethical concerns with this paper?

No

Have you any concerns about statistical analyses in this paper?

No

Recommendation?

Accept with minor revision

Comments to the Author(s)

Comments on „Are post-error adjustments influenced by beliefs in free will? A direct replication of Rigoni, Wilquin, Brass, and Burle, 2013“.

The motivation for performing the replication study is reasonable; the methods and planned analyses – including the described deviations from the original procedure – seem ok to me.

I have only a couple of relatively minor comments on the proposal / manuscript that the authors might wish to consider when revising their proposal / manuscript:

1. Language. The authors use different terms for describing the parts of their (planned) experiment. In particular, the authors speak of “test moments” in the introduction, and later they speak of “test parts” in the procedure section. This is a bit confusing, and I recommend using consistent wording. I also found “test moment” a rather unusual term; what about “test phase”?
2. Procedure. The authors are planning to run the two conditions consecutively: first the control condition, and then the anti-free will condition. I assume the authors want to prevent carry-over effects from the anti-free will manipulation to the control condition, when the former condition would occur before the latter, and this makes sense to me. However, the fixed order means that participants will always have more practice (or more fatigue) in the experimental (anti free-will condition) as compared to the control condition. I encourage the authors to add a statement concerning whether anything is known about the effects of practice and/or fatigue on post-error slowing? More generally: Do post-error slowing effects – for whatever reason – vary between earlier and later parts of an experiment? If there are no published data on this issue, the authors might conduct a corresponding analysis on the data from their pilot study, and add a statement on the results.
3. Data analysis. The authors might justify their preference for two-tailed tests over one-tailed tests, despite having directional hypotheses.

Review form: Reviewer 2

Do you have any ethical concerns with this paper?

No

Have you any concerns about statistical analyses in this paper?

No

Recommendation?

Major revision

Comments to the Author(s)

It seems most relevant to try and replicate the Rigoni ea findings, and I applaud this effort. It will be quire interesting to learn whether the Rigoni effect can be replicated, or was rather a false positive.

Should the results of the replication study turn out to differ from the original study, this could potentially be attributed to a number of factors, including

- larger / more robust sample (reducing the chances of an false positive outcomes)
- differences in procedures (e.g., online instead of hands-on; 1 session instead of 2 with a week in between)
- differences in measures (e.g., FWI instead of FAD; different method of calculating and analyzing RT effects)

Hence, this is far from an exact replication. Although we'd want eventual differences, if any, to depend solely on sample size, each and every of the other differences might account for different outcomes as well. For instance, if participants are just slightly more lax and slow in responding

online compared to in the lab, then the PES effect may go away just for that reason, leaving the authors empty-handed.

The issue of measures should not be too devastating in the case of FWI vs FAD+, since the field apparently has decided that the former is more sensitive than the latter; this should work in favor of the chances to replicate. The issue of measures should further be easy to address in case of the RT measures of PES, which could well be analysed in both the traditional way and the new, now preferred way (perhaps this is what the authors already intended, but this is not entirely clear from the stage-1 manuscript).

The fact that the authors already piloted their new online task, and that it did produce typical PES effects is also reassuring.

But other potential confounds remain (such as whether the effects of reading a piece of text are as profound when read online vs in the lab).

In principle, I don't see any immediately compelling reason why this replication study should be conducted online, except that it's probably easier to do larger samples online, and less work to do 1 session instead of 2. But it certainly doesn't impossible, or even superproportionally challenging, to run a study like this in the lab.

One way out would be to run another (preregistered) pilot to verify that the free will manipulation *does* work in a single on-line session. As the authors state, their meta-analysis indicates robust effects of the free will manipulation on certain questionnaires; if they can demonstrate that similar effects can be observed in a single on-line session, then the reader might (and this reviewer would) be reassured of the potential power of the manipulation when implemented online. If, however, the outcomes of such a pilot fail to demonstrate such power, then clearly the replication study would have to resort to testing procedures that more closely replicate the lab procedures of the original study.

Decision letter (RSOS-200664.R0)

Dear Ms Eben,

The Editors assigned to your Stage 1 Replication submission ("Are post-error adjustments influenced by beliefs in free will? A direct replication of Rigoni, Wilquin, Brass and Burle, 2013") have now received comments from reviewers. We would like you to revise your paper in accordance with the referee and editors suggestions which can be found below (not including confidential reports to the Editor). Please note this decision does not guarantee eventual acceptance.

When submitting your revised manuscript, you must respond to the comments made by the referees and upload a file "Response to Referees" in the "File Upload" step. Please use this to

document how you have responded to the comments, and the adjustments you have made. In order to expedite the processing of the revised manuscript, please be as specific as possible in your response.

Once again, thank you for submitting your manuscript to Royal Society Open Science and we look forward to receiving your revision. If you have any questions at all, please do not hesitate to get in touch. Full author guidelines may be found at <https://royalsocietypublishing.org/rsos/replication-studies#AuthorsGuidance>.

Kind regards,

on behalf of Professor Chris Chambers (Registered Reports Editor, Royal Society Open Science)
openscience@royalsociety.org

Associate Editor Comments to Author (Professor Chris Chambers):

Two expert reviewers have now assessed the manuscript. Both are broadly positive while also raising a number of concerns that will need to be carefully addressed to achieve Stage 1 acceptance. The most substantive issue is the deviation from the original methods, which Reviewer 2 judges to fail to meet primary criterion #2 (that replications must be as close as possible). There is room for discretion in this judgment, as some minor variation is often unavoidable or irrelevant to theory, but it may also be that further empirical work is needed to more closely align the methods of replication with the original research. I will leave the authors to decide exactly how to respond, but addressing this concern to the satisfaction of Reviewer 2 will be key to achieving IPA, as well as the other issues raised by Reviewer 1 and 2.

Reviewer Comments to Author:

Reviewer: 1
 Comments to the Author(s)

Comments on „Are post-error adjustments influenced by beliefs in free will? A direct replication of Rigoni, Wilquin, Brass, and Burle, 2013“.

The motivation for performing the replication study is reasonable; the methods and planned analyses – including the described deviations from the original procedure – seem ok to me.

I have only a couple of relatively minor comments on the proposal / manuscript that the authors might wish to consider when revising their proposal / manuscript:

1. Language. The authors use different terms for describing the parts of their (planned) experiment. In particular, the authors speak of “test moments” in the introduction, and later they speak of “test parts” in the procedure section. This is a bit confusing, and I recommend using consistent wording. I also found “test moment” a rather unusual term; what about “test phase”?
2. Procedure. The authors are planning to run the two conditions consecutively: first the control condition, and then the anti-free will condition. I assume the authors want to prevent carry-over effects from the anti-free will manipulation to the control condition, when the former condition would occur before the latter, and this makes sense to me. However, the fixed order means that participants will always have more practice (or more fatigue) in the experimental (anti free-will

condition) as compared to the control condition. I encourage the authors to add a statement concerning whether anything is known about the effects of practice and/or fatigue on post-error slowing? More generally: Do post-error slowing effects – for whatever reason – vary between earlier and later parts of an experiment? If there are no published data on this issue, the authors might conduct a corresponding analysis on the data from their pilot study, and add a statement on the results.

3. Data analysis. The authors might justify their preference for two-tailed tests over one-tailed tests, despite having directional hypotheses.

Reviewer: 2

Comments to the Author(s)

It seems most relevant to try and replicate the Rigoni et al findings, and I applaud this effort. It will be quite interesting to learn whether the Rigoni effect can be replicated, or was rather a false positive.

Should the results of the replication study turn out to differ from the original study, this could potentially be attributed to a number of factors, including

- larger / more robust sample (reducing the chances of a false positive outcome)
- differences in procedures (e.g., online instead of hands-on; 1 session instead of 2 with a week in between)
- differences in measures (e.g., FWI instead of FAD; different method of calculating and analyzing RT effects)

Hence, this is far from an exact replication. Although we'd want eventual differences, if any, to depend solely on sample size, each and every of the other differences might account for different outcomes as well. For instance, if participants are just slightly more lax and slow in responding online compared to in the lab, then the PES effect may go away just for that reason, leaving the authors empty-handed.

The issue of measures should not be too devastating in the case of FWI vs FAD+, since the field apparently has decided that the former is more sensitive than the latter; this should work in favor of the chances to replicate. The issue of measures should further be easy to address in case of the RT measures of PES, which could well be analysed in both the traditional way and the new, now preferred way (perhaps this is what the authors already intended, but this is not entirely clear from the stage-1 manuscript).

The fact that the authors already piloted their new online task, and that it did produce typical PES effects is also reassuring.

But other potential confounds remain (such as whether the effects of reading a piece of text are as profound when read online vs in the lab).

In principle, I don't see any immediately compelling reason why this replication study should be conducted online, except that it's probably easier to do larger samples online, and less work to do 1 session instead of 2. But it certainly doesn't seem impossible, or even superproportionally challenging, to run a study like this in the lab.

One way out would be to run another (preregistered) pilot to verify that the free will manipulation *does* work in a single on-line session. As the authors state, their meta-analysis indicates robust effects of the free will manipulation on certain questionnaires; if they can demonstrate that similar effects can be observed in a single on-line session, then the reader might (and this reviewer would) be reassured of the potential power of the manipulation when implemented online. If, however, the outcomes of such a pilot fail to demonstrate such power,

then clearly the replication study would have to resort to testing procedures that more closely replicate the lab procedures of the original study.

Author's Response to Decision Letter for (RSOS-200664)

See Appendix A.

RSOS-200664.R1 (Revision)

Review form: Reviewer 2

Do you have any ethical concerns with this paper?

No

Have you any concerns about statistical analyses in this paper?

No

Recommendation?

Accept in principle

Comments to the Author(s)

My concerns were handled satisfactorily.

Decision letter (RSOS-200664.R1)

Dear Ms Eben

On behalf of the Editor, I am pleased to inform you that your Manuscript RSOS-200664.R1 entitled "Are post-error adjustments influenced by beliefs in free will? A direct replication of Rigoni, Wilquin, Brass and Burle, 2013" has been accepted in principle for publication in Royal Society Open Science.

You may now progress to Stage 2 and complete the study as approved.

Please note that you must now register your approved protocol on the Open Science Framework (<https://osf.io/rr>), using the 'Submit your approved Registered Report' option and then the 'Registered Report Protocol Preregistration' option. Please use the Registered Report option even though your article is being accepted as a Stage 1 Replication. Further into the registration process, in the Journal Title field enter 'Royal Society Open Science (Replication article type, Results-Blind track)'. Please note that a time-stamped, independent registration of the protocol is mandatory under journal policy, and manuscripts that do not conform to this requirement cannot be considered at Stage 2. The protocol should be registered unchanged from its current approved

state. Please include a URL to the protocol in your Stage 2 manuscript, and because you submitted via the Results-Blind track please note in the manuscript that the pre-registration was performed after data analysis (e.g. 'This article received results-blind in-principle acceptance (IPA) at Royal Society Open Science. Following IPA, the accepted Stage 1 version of the manuscript, not including results and discussion, was preregistered on the OSF (URL). This preregistration was performed after data analysis.')

Following completion of your study, we invite you to resubmit your paper for peer review as a Stage 2 Replication. Please note that your manuscript can still be rejected for publication at Stage 2 if the Editors consider any of the following conditions to be met:

- The Introduction and methods deviated from the approved Stage 1 submission.
- The authors' conclusions were not considered justified given the data.

We encourage you to read the complete guidelines for authors concerning Stage 2 submissions at: <https://royalsocietypublishing.org/rsos/replication-studies#AuthorsGuidance>. Please especially note the requirements for data sharing and that withdrawing your manuscript will result in publication of a Withdrawn Registration.

Once again, thank you for submitting your manuscript to Royal Society Open Science and I look forward to receiving your Stage 2 submission. If you have any questions at all, please do not hesitate to get in touch. We look forward to hearing from you shortly with the anticipated submission date for your stage two manuscript.

Kind regards,
Andrew Dunn
Royal Society Open Science
openscience@royalsociety.org

on behalf of Chris Chambers (Registered Reports Editor, Royal Society Open Science)
openscience@royalsociety.org

Reviewers' comments to Author:
Reviewer: 2

Comments to the Author(s)
My concerns were handled satisfactorily.

Author's Response to Decision Letter for (RSOS-200664.R1)

See Appendix B.

RSOS-200664.R2 (Revision)

Review form: Reviewer 1

Do you have any ethical concerns with this paper?

No

Have you any concerns about statistical analyses in this paper?

No

Recommendation?

Accept as is

Comments to the Author(s)

This is the stage-2 part of a pre-registered replication report on the possible effects of the belief in free will on post-error slowing as a behavioral measure of cognitive control. I have already seen the stage-1 version of the project, and I think the present submission is an acceptable stage-2 successor.

Review form: Reviewer 2

Do you have any ethical concerns with this paper?

No

Have you any concerns about statistical analyses in this paper?

No

Recommendation?

Accept as is

Comments to the Author(s)

Just a final suggestion: the way in which I would interpret the title of the paper is that this study replicates the findings of the previous one. However, as the authors state, this study is a failed replication. Hence, I would suggest "direct failure to replicate" for the title instead of "direct replication".

Decision letter (RSOS-200664.R2)

Dear Ms Eben:

It is a pleasure to accept your manuscript entitled "Are post-error adjustments influenced by beliefs in free will? A direct replication of Rigoni, Wilquin, Brass and Burle, 2013" in its current form for publication in Royal Society Open Science. The comments of the reviewer(s) who reviewed your manuscript are included at the foot of this letter.

Kind regards,
Professor Chris Chambers
Royal Society Open Science
openscience@royalsociety.org

on behalf of Professor Chris Chambers (Registered Reports Editor, Royal Society Open Science)
openscience@royalsociety.org

Associate Editor Comments to Author (Professor Chris Chambers):

Associate Editor: 1

Comments to the Author:

Both reviewers recommend full acceptance without the need for further revision, and I having read the manuscript myself I am happy to say I am in full agreement. Reviewer 2 suggests that the title could be changed to avoid potentially misleading readers to think that the results were a non-replication rather than a replication. I agree that this would be less ambiguous, but equally the term "direct replication" is also often used in psychology to describe a study type rather than a study outcome. For this reason, I'm happy for the authors to decide about this, and if they wish to change the title then to avoid another round of submission, they are welcome to do so at the proof stage.

Reviewer(s)' Comments to Author:

Reviewer: 1

Comments to the Author(s)

This is the stage-2 part of a pre-registered replication report on the possible effects of the belief in free will on post-error slowing as a behavioral measure of cognitive control. I have already seen the stage-1 version of the project, and I think the present submission is an acceptable stage-2 successor.

Reviewer: 2

Comments to the Author(s)

Just a final suggestion: the way in which I would interpret the title of the paper is that this study replicates the findings of the previous one. However, as the authors state, this study is a failed replication. Hence, I would suggest "direct failure to replicate" for the title instead of "direct replication".

Appendix A

Dear Dr. Chambers,

This letter accompanies the revision of our manuscript entitled “Are post-error adjustments influenced by beliefs in free will? A direct replication of Rigoni, Wilquin, Brass and Burle, 2013”. We would like to thank you and the reviewers for the positive reception and insightful feedback.

Reviewer 2 had some concerns about the differences with the original study, and in particular, our decision to run the study online and in a single session (instead of two lab sessions as in the original study). We did not stress clearly enough that recently published studies (Genschow et al. 2017, Moynihan et al. 2019) and a meta-analysis (in preparation) conducted by some of the authors of the present manuscript, indicate that the Crick text used to manipulate belief in free will also works in online studies (with a single session). There might be a small numerical difference in effect size, but the meta-analysis (based on 20+ lab-based and 20+ online studies) indicates that this difference is not significant ($p = .21$). As discussed below, we have added this information in our revised manuscript. We therefore assume that an additional pilot is not necessary to prove that this manipulation can work in an online setting.

Note that we opted to run the study online, as a between-subject design is required to test our hypothesis and effects of free-will beliefs on cognitive measures are likely to be small (again, based on the meta-analysis). Thus, hundreds of participants might be required to determine if the effect is present or not. For practical reasons, this is not feasible in a traditional lab context.

Finally, we would like to mention that one of our authors (Marcel Brass) was also author of the original study (and related follow-up work), and considers this study to be a valid replication attempt.

A more complete and detailed response to the comments can be found below (and where appropriate, a reference to the respective changes in the manuscript is given). We believe that the comments helped us to further improve the paper. We thank you for considering our manuscript for publication and look forward to hearing from you in due course.

Yours sincerely,
Charlotte Eben
Zhang Chen
Emiel Cracco
Marcel Brass
Joël Billieux
Frederick Verbruggen

Reviewer: 1

1. Language. The authors use different terms for describing the parts of their (planned) experiment. In particular, the authors speak of “test moments” in the introduction, and later they speak of “test parts” in the procedure section. This is a bit confusing, and I recommend using consistent wording. I also found “test moment” a rather unusual term; what about “test phase”?

Reply: Thank you for pointing this out. We have corrected this and consistently use the term 'test phase' in the revised version of the manuscript.

2. Procedure. The authors are planning to run the two conditions consecutively: first the control condition, and then the anti-free will condition. I assume the authors want to prevent carry-over effects from the anti-free will manipulation to the control condition, when the former condition would occur before the latter, and this makes sense to me. However, the fixed order means that participants will always have more practice (or more fatigue) in the experimental (anti free-will condition) as compared to the control condition. I encourage the authors to add a statement concerning whether anything is known about the effects of practice and/or fatigue on post-error slowing? More generally: Do post-error slowing effects – for whatever reason – vary between earlier and later parts of an experiment? If there are no published data on this issue, the authors might conduct a corresponding analysis on the data from their pilot study, and add a statement on the results.

Reply: We apologize for the confusion. Our manipulation is a between-subjects manipulation (as in the original study). However, to ensure that we had an equal number of participants in both groups, we had to create two 'experiments' on prolific (one per group). To ensure that participants could not participate in both 'experiments', those who had taken part in one experiment (group) were not eligible for the other experiment (group). But exclusion based on another experiment (or in our case, group) on prolific is only possible when the experiment is finished/closed. Therefore the two groups (control vs. anti-free-will) were tested consecutively. We adjusted the wording to avoid confusion with the reader which can be found on p.4-5..

3. Data analysis. The authors might justify their preference for two-tailed tests over one-tailed tests, despite having directional hypotheses.

Reply: It is true that we are trying to replicate a specific effect. However, we had some concerns about the sample size (and the direction of the effect) of the original study, and we were not sure whether we would be able to replicate the original finding. Therefore, we used two-tailed t-tests because (in theory) the effect could also go in the opposite direction, and such a finding could still be theoretically meaningful. We added this reasoning to the ms on p. 6-7 in the data analysis section.

Reviewer: 2

Should the results of the replication study turn out to differ from the original study, this could potentially be attributed to a number of factors, including - larger / more robust sample (reducing the chances of an false positive outcomes) - differences in procedures (e.g., online instead of hands-on; 1 session instead of 2 with a week in between) - differences in measures (e.g., FWI instead of FAD; different method of calculating and analyzing RT effects) Hence, this is far from an exact replication. Although we'd want eventual differences, if any, to depend solely on sample size, each and every of the other differences might account for different outcomes as well.

For instance, if participants are just slightly more lax and slow in responding online compared to in the lab, then the PES effect may go away just for that reason, leaving the authors empty-handed.

Reply: As acknowledged by the reviewer, we piloted our online Simon task and observed the typical PES effect. In fact, we observed PES in several other online studies that were recently conducted in our lab. Therefore, we are confident that we can replicate the basic PES effect in an online setting.

The issue of measures should not be too devastating in the case of FWI vs FAD+, since the field apparently has decided that the former is more sensitive than the latter; this should work in favor of the chances to replicate. The issue of measures should further be easy to address in case of the RT measures of PES, which could well be analysed in both the traditional way and the new, now preferred way (perhaps this is what the authors already intended, but this is not entirely clear from the stage-1 manuscript).

Reply: We have now emphasized in the data processing section that we will use both methods to analyse PES.

*But other potential confounds remain (such as whether the effects of reading a piece of text are as profound when read online vs in the lab). In principle, I don't see any immediately compelling reason why this replication study should be conducted online, except that it's probably easier to do larger samples online, and less work to do 1 session instead of 2. But it certainly doesn't impossible, or even superproportionally challenging, to run a study like this in the lab. One way out would be to run another (preregistered) pilot to verify that the free will manipulation *does* work in a single on-line session. As the authors state, their meta-analysis indicates robust effects of the free will manipulation on certain questionnaires; if they can demonstrate that similar effects can be observed in a single on-line session, then the reader might (and this reviewer would) be reassured of the potential power of the manipulation when implemented online. If, however, the outcomes of such a pilot fail to demonstrate such power, then clearly the replication study would have to resort to testing procedures that more closely replicate the lab procedures of the original study.*

Reply: Thank you for pointing this out. We were indeed not clear enough regarding existing evidence of the effectiveness of the Crick manipulation in

online experiments.

To our knowledge there are at least two published studies (Genschow et al. 2017, Moynihan et al. 2019) that used the Crick text in a single-session online experiment with large samples, and both studies did find a difference between the anti-free will group and the control group in their manipulation checks. Here we will include the same manipulation checks.

Moreover, the meta-analysis (Genschow et al. in prep) mentioned above included these two studies and other unpublished studies to compare lab and online results regarding this specific Crick-text manipulation. The meta-analysis found an effect in the lab studies, ($g = 0.21$, $SE = 0.04$, $p < 0.001$, based on 25 effect sizes from 24 experiments), as well as an effect in the online studies ($g = 0.13$, $SE = 0.05$, $p = 0.012$, based on 22 effect sizes from 20 experiments, including the two studies already mentioned). This numerical difference in effect size was not significant ($p = .21$). We added this clarification to the method section on p. 5.

As the meta-analysis indicates that the effect is likely to be small (at best), regardless of whether we would run the experiment in the lab or online, we decided to opt for an online study. After all, the free-will manipulation has to be between-subjects (to avoid carry-over effects). Given the expected small effect size, this could require testing hundreds of participants (even when a Bayesian stopping rule is used). Therefore, we considered it not feasible to run this study in the lab (at least not within a reasonable time frame).

As a last point, one of our authors (Marcel Brass) was also author of the original study (and related follow-up work) and considers this study as a valid and 'fair' replication attempt.

Appendix B

Dear Dr. Chambers,

We would like to submit our manuscript “Are post-error adjustments influenced by beliefs in free will? A direct replication of Rigoni, Wilquin, Brass and Burle, 2013” to Royal Society Open Science as a Stage 2 – Replication Study (IPA received on 15/06/2020).

We would like to emphasize two things here. First of all, this study has been pre-registered prior to submission (<https://osf.io/kmpdh>). Following an email conversation with Dr. Andrew Dunn (19/06/2020), we did not register our study plan a second time after receiving IPA as the study plan was accepted as it was.

Second, we changed two things in the Stage 1 accepted version, which we also highlighted in the additional PDF. We added the following sentence on p. 7: “In contrast to the pre-registration, in which we said we would perform all analyses in R, we used JASP (version 0.13.1.0) to perform the Bayesian ANOVA. We report the inclusion BF_{10} of the Bayesian ANOVAs (calculated in JASP) in Tables 1-7. The BF_{10} of the Bayesian t-tests (calculated in R) are reported in the main text. An inclusion BF_{10} of B means the data are about B times more likely under the models that include this effect, compared to the models that do not include this effect.”

We also had to correct the effect size of the pilot study on p. 5: “We tested this Simon task procedure in a pilot study (N=40) and observed the expected PES (Hedge’s $g_{av} = 1.6$.” In the accepted stage 1 version we had a wrong effect size of 3.2.

The present manuscript is original, has not been published elsewhere and is not under consideration for publication elsewhere. In addition to the pre-registered study plan, the corresponding data, experimental material and analyses scripts were archived on OSF (<https://osf.io/3nxpg/>).

Yours sincerely,
Charlotte Eben
Zhang Chen
Emiel Cracco
Marcel Brass
Joël Billieux
Frederick Verbruggen